# Gepants for Acute and Preventive Migraine Treatment: A Narrative Review

**DOI:** 10.3390/brainsci12121612

**Published:** 2022-11-24

**Authors:** Jamir Pitton Rissardo, Ana Letícia Fornari Caprara

**Affiliations:** Medicine Department, Federal University of Santa Maria, Santa Maria 97105-900, Brazil

**Keywords:** rimegepant, ubrogepant, atogepant, zevagepant, migraine

## Abstract

Calcitonin gene-related peptide (CGRP) antagonists are a class of medications that act as antagonists of the CGRP receptor or ligand. They can be divided into monoclonal antibodies and non-peptide small molecules, also known as gepants. CGRP antagonists were the first oral agents specifically designed to prevent migraines. The second generation of gepants includes rimegepant (BHV-3000, BMS-927711), ubrogepant (MK-1602), and atogepant (AGN-241689, MK-8031). Zavegepant (BHV-3500, BMS-742413) belongs to the third generation of gepants characterized by different administration routes. The chemical and pharmacological properties of this new generation of gepants were calculated. The clinical trials showed that the new generation of CGRP antagonists is effective for the acute and/or preventive treatment of migraines. No increased mortality risks were observed to be associated with the second- and third-generation gepants. Moreover, the majority of the serious adverse events reported probably occurred unrelated to the medications. Interesting facts about gepants were highlighted, such as potency, hepatotoxicity, concomitant use with monoclonal antibodies targeting the CGRP, comparative analysis with triptans, and the “acute and preventive” treatment of migraine. Further studies should include an elderly population and compare the medications inside this class and with triptans. There are still concerns regarding the long-term side effects of these medications, such as chronic vascular hemodynamic impairment. Meanwhile, careful pharmacovigilance and safety monitoring should be performed in the clinical practice use of gepants.

## 1. Introduction

Migraine is a disabling disorder that affects approximately 15% of the global population [1]. In the United States (US), migraine is encountered among 70 million people and is the second most prevalent neurological disorder. The incidence of migraine headaches was 1722 cases per 100,000 people in 2017. The “Burden of Neurological Disorders Across the US From 1990–2017” estimated that migraine is the third most burdensome neurological disorder in terms of disability-adjusted life-years [2]. Migraine also most commonly occurs at productive age, in which the frequency of migraine headache days correlate with increased disability, leading to a decreased quality of life involving negative psychological impacts on performances within social and familial contexts [3]. Moreover, the estimated mean cost for migraine-related care per outpatient visit is $139.88, per emergency room visit is $775.09, and per inpatient hospitalization is $7317.07 [4]. Therefore, this disease affects an important percentage of the US population and represents a significant burden to the health care system.

The pathophysiology of migraine remains poorly understood. However, some specific vasoactive substances and neurotransmitters probably play a role in the mechanism of neurovascular and cortical spreading depression [5]. Calcitonin gene-related peptide (CGRP), neurokinin A, nitric oxide, and substance P are released with perivascular nerve activity [6]. They likely interact with the blood vessel wall, causing dilation, protein extravasation, and sterile inflammation [7]. The stimulation of the trigeminocervical complex by these chemical substances is relayed to the thalamus and cortex, both of which register the pain [8]. Interestingly, it was observed that the patients with migraine have reduced functioning of endogenous pain-control pathways that gate the pain [9]. Thus, the headache pain process requires not only the activation of nociceptors of pain-producing intracranial structures but also an impairment in its neuromodulation.

In 1992, the Food and Drug Administration (FDA) approved sumatriptan succinate, trade name Imitrex^®^, for the acute treatment of migraine with or without aura in adults [10]. The development of this drug was a benchmark for the acute therapy and pathophysiology of migraine. It was first assumed that triptans possibly relieved pain by direct vasoconstriction and regulation of trigeminal inflammatory transmitter release [11]. Over the last decades, preventive migraine therapy relied on medications not specifically developed for migraine management. In this way, individuals with migraine were burdened by poor long-term adherence due to adverse events and often inadequate effectiveness [12]. It is worth mentioning that migraine often comes associated with different comorbidities, which may further restrict the range of therapeutic choices [13]. Therefore, the advent and approval of new drugs, such as antagonists of the CGRP receptors particularly designed for migraines, was a remarkable development in the acute and preventive treatment of migraine (Appendix A).

## 2. CGRP and GEPANTS

In 1982, the CGRP structure was discovered by Amara et al [14]. They noted that this compound was mainly synthesized in the hypothalamus, suggesting a hormonal effect for CGRP. A few years later, it was observed that CGRP-specific mRNA could be encountered throughout the central nervous system and in the peripheral nervous system, including the visceral motor functions mediated by the vagus nerve [15]. Rabbit models showed that intradermal injection of CGRP induced microvascular dilatation, resulting in increased blood flow [16]. Interestingly, CGRP is the most potent peripheral and cerebral vasodilator molecule ever discovered [17].

The potential role of the neurovascular system in migraine was hypothesized. In 1988, the first preclinical studies in cat models showed a potential association between CGRP and migraine [18]. It was observed that the irritation of the trigeminovascular system released CGRP in the extra-cerebral circulation. Two years later, Goadsby et al demonstrated dynamic changes in the concentrations of CGRP during migraine attacks in the external jugular but not in the cubital fossa blood [19]. Lassen et al infused CGRP in individuals with migraine to elucidate the mechanism of CGRP and migraine. They observed that intravenous administration of CGRP caused migraine in migraineurs [20].

CGRP receptors are transmembrane G protein-coupled receptors produced by the association between calcitonin receptor-like receptors (CALCRL) and a receptor activity-modifying protein (RAMP1) [21]. CGRP-expressing neurons are either C-fiber or Aδ-fiber [22]. Interestingly, more than thirty percent of trigeminal ganglion neurons exhibit CGRP receptors [23]. CGRP antagonists are a class of medications that act as antagonists of the CGRP receptor or ligand. They can be divided into monoclonal antibodies and non-peptide small molecules, also known as gepants (Table 1).

The first generation of gepants raised concerns about liver toxicity and poor oral availability [24]. These drugs were BI 44370 TA (BI 44370), MK-3207, olcegepant (BIBN-4096BS), and telcagepant (MK-0974). Thus, the studies in the early 2010s were terminated, and the results were probably overlooked [25]. After almost a decade, the pharmaceutical industry retrieved and reanalyzed the results with rimegepant. They observed that rimegepant was effective, compared to the placebo and sumatriptan 100 mg for managing acute migraine. Furthermore, rimegepant did not show significant abnormalities in liver enzymes, even with participants receiving a single dose (four capsules of 150 mg each) of rimegepant 600 mg orally within 45 days of randomization once they experienced a migraine headache of moderate to severe intensity [26].

The favorable results of rimegepant’s first clinical trials were followed by ongoing studies with other gepants. In this way, the second generation of gepants emerged, and it includes rimegepant (BHV-3000, BMS-927711), ubrogepant (MK-1602), and atogepant (AGN-241689, MK-8031). These drugs revealed a low number of side effects, with excellent bioavailability due to molecular modifications. In December 2019, the FDA approved ubrogepant (Ubrelvy^®^) for the acute treatment of migraine with or without aura in adults. This drug was the first oral gepant approved by the FDA [27]. It is worth mentioning that the second generation has an affinity for different receptors besides the CGRP receptors [28]. The first-generation affinity for the human CGRP receptors is also higher than the second generation of gepants [29]. 

Rimegepant has a unique drug formulation, characterized as an oral dissolving tablet that disperses in the mouth, facilitating medication delivery and adhesion [30]. In addition, this formulation improves patients’ compliance to the medication due to decreased triggers such as nausea and vomiting commonly associated with migraine [31].

The third generation of gepants is characterized by different routes of administration. In this context, zavegepant (BHV-3500, BMS-742413) is under study for different types of routes, including subcutaneous and intranasal, due to the pharmacological properties of this molecule [32].

We calculated the chemical and pharmacological properties of the second and third-generation gepants using the SwissADME tool (Figure 1). These properties can help identify compounds suitable for oral use [33]. All the parameters analyzed were within the normal range, except for the molecular weights that were slightly higher than those desired for oral molecules, which can reduce oral bioavailability. Refer to the Appendix A for a complete description of physicochemical descriptors and pharmacokinetic characteristics of rimegepant, ubrogepant, atogepant, and zavegepant (Appendix A). It is important to mention that some pharmacological features of this new group of medications still require further investigation.

## 3. Clinical Trials Related to Gepants

The results of the clinical trials related to the gepants indicated for acute migraine therapy are summarized in Table 2 [31,34,35,36,37,38,39,40]. No increased mortality risks were observed to be associated with the second- and third-generation gepants. Moreover, the majority of the serious adverse events reported probably occurred unrelated to the medications.

A table was uploaded as Appendix A with all the clinical trials related to rimegepant, ubrogepant, atogepant, and zavegepant; it was registered at “ClinicalTrials.gov” (Appendix A). A figure is provided with the conditions assessed in these clinical trials (Figure 2). Colors are graded according to possible mechanisms related to CGRP, such as vascular and inflammatory pathways.

### 3.1. Rimegepant

The first clinical trial of rimegepant was performed in 2012 with the objective of dose-ranging. The NCT01430442 demonstrated that the best dose of rimegepant is 75 to 150 mg PO once a day. Interestingly, there was a low number of side effects when compared to other clinical trials with rimegepant. The other three studies, NCT03235479, NCT03237845, and NCT03461757, showed the efficacy of rimegepant 75 mg to placebo [31,34,35,36]. There was also no statistical difference between the adverse events of the rimegepant and placebo groups. The serious adverse effects reported in the studies with rimegepant were acute respiratory failure, back pain, pneumonia, post-lumbar puncture syndrome, pulmonary embolism, and stress cardiomyopathy. 

### 3.2. Ubrogepant

Ubrogepant studies had the highest dropout rate compared to rimegepant, zavegepant, and atogepant [37,38,39]. However, the meta-analysis assessing the causes of withdrawal of the studies with gepants, lasmiditan, and triptans did not show any relation to the adverse effects. The ubrogepant clinical trials showed serious adverse events, including myoclonus, pericardial effusion, seizure, and spontaneous abortion. We hypothesized that the more common central nervous system side effects related to ubrogepant were probably associated with the affinity to the human CGRP receptor. A study using docking and molecular dynamics simulations showed that ubrogepant bound more strongly to the human CGRP receptor than rimegepant.

### 3.3. Zavegepant

The preliminary results of the zavegepant did not show any serious adverse events. The most common adverse effects were dysgeusia and nasal discomfort [40]. There were no abnormalities in liver enzymes during the study. This can be partially explained by the route of administration and the bioavailability of zavegepant. Further studies, including oral administration, are needed to better characterize side effects and pharmacological properties.

### 3.4. Atogepant

Atogepant was the first gepant developed exclusively as a preventive treatment for migraine (Table 3) [41,42,43]. Its affinity at the CGRP receptor in humans is higher than that of ubrogepant. The atogepant also has an affinity for the AMY1 receptors [44]. The combination with onabotulinumtoxin A effectively reduced sensitization and cortical spreading depression in an animal model [45]. Among the gepants already approved for managing migraine, atogepant has the highest percentage of side effects, including serious adverse events.

## 4. Discussion

The advent of new medications comes with uncertainties and concerns about possible adverse events, efficacy, and costs. Herein we would like to discuss some interesting facts revealed in recent clinical trials and systematic reviews.

### 4.1. Gepants Efficacy and Potency

The comparison of efficacies among different types of drugs inside the same class of medication is important. It contributes to the design of specific molecules with tailored properties according to their main uses. There are no comparative trials assessing rimegepant and ubrogepant. In this way, two recent network meta-analyses showed slightly different results regarding the most efficacious drug among rimegepant, ubrogepant, and lasmiditan, although differences in design limited detailed comparisons between these studies.

Johnston et al studied rimegepant 75 mg (orally dissolving tablet), ubrogepant 25 mg, 50 mg, 100 mg (oral), and lasmiditan 50 mg, 100 mg, and 200 mg (oral). The efficacy outcomes included sustained pain freedom and pain relief 2–24 h post-dose. They also added safety outcomes, which were drowsiness and dizziness. Rimegepant was found to be more efficacious than the placebo and lower doses of lasmiditan and ubrogepant concerning sustained pain freedom. It is relevant to mention that higher doses of lasmiditan were associated with increased dizziness [46].

Polavieja et al performed a similar collection of data, including the same drugs and doses, but a different methodology regarding statistical analysis was used. They included new evidence for lasmiditan from the registration studies CENTURION and MONONOFU. Polavieja and colleagues noted that lasmiditan was significantly associated with central nervous system side effects. However, the most efficacious drug was lasmiditan 100 and 200 mg, compared to the other drugs and doses [47].

Refer to the Appendix A for a detailed description of the four network meta-analyses comparing the efficacies of lasmiditan, rimegepant, and ubrogepant published in the previous literatures (Appendix A) [46,47,48,49].

### 4.2. Gepants Hepatotoxicity

Telcagepant, also known as MK-0974, is a first-generation CGRP receptor antagonist developed by Merck & Co. It was an effective drug with similar potency to commonly used triptans for managing acute migraine [50,51]. However, in a Phase IIa clinical trial (NCT00797667), two patients in the telcagepant group had significant symptomatic abnormalities in serum transaminases. A substantial increase in transaminases was also observed in 13 individuals taking telcagepant [52]. Therefore, the studies with telcagepant and migraine headaches were terminated by FDA coining Hy’s law [53].

The development of the second-generation CGRP antagonists brought doubts about possible hepatotoxicity. In this context, clinical trials with these compounds carefully monitored transaminases, and none of the patients developed liver function abnormalities with these drugs. 

Woodhead et al compared the liver safety profile of the second- and third-generation CGRP receptor antagonists to the hepatotoxic CGRP inhibitor telcagepant using quantitative systems toxicology modeling (DILIsym v6A) [54]. They measured the potential for each compound to inhibit bile acid transporters, produce oxidative stress, and cause mitochondrial dysfunction. The drug-induced liver injury (DILI) quantitative systems toxicology model predicted clinical elevations of liver enzymes and bilirubin for telcagepant. On the other hand, DILIsym predicted that the new-generation compounds would be significantly less likely to cause DILI than telcagepant. In this way, the ascending order of hepatotoxicity [oral dosing protocol] is zavegepant [750 mg oral QD, 25 days, 25 total doses] (0%), rimegepant [75 mg QD, daily dosing for 25 days, 25 total doses] (1%), atogepant [600 mg BID, 12 weeks] (10.2%), ubrogepant [1000 mg QD, 25 days] (11.6%), and telcagepant [280 mg BID, 12 weeks] (76.1%). On the other hand, these doses are not commonly prescribed in clinical practice.

### 4.3. Gepants and Monoclonal Antibodies Targeting the CGRP

Erenumab, trade name Aimovig^®^, was the first medication approved in the United States to prevent migraines in adults. The mechanism of this drug consisted of the antagonism of calcitonin gene-related peptide (CGRP) receptors [55]. Millions of individuals were already using these medications for the chronic treatment of migraine when gepants came to the market. Most people who take monoclonal antibodies targeting the CGRP continue to experience breakthrough attacks that require acute treatment [56]. Therefore, studies of interactions between the gepants for the acute management of migraine and monoclonal antibodies for preventive treatment are needed.

Berman and colleagues assessed the safety of rimegepant plus erenumab, fremanezumab, and galcanezumab [57]. They observed that rimegepant, when used as an oral acute treatment in patients receiving CGRP monoclonal antibodies as a prophylactic treatment, was well tolerated, and none of the patients had serious adverse events. However, this study included a small number of patients. Thus, multicenter studies involving a more significant number of patients with migraine are required.

Jakate et al evaluate the impact of erenumab and galcanezumab on the pharmacokinetic profile, safety, and tolerability of ubrogepant [58]. There were no significant changes in the pharmacokinetics of ubrogepant, nor were safety issues identified. Nevertheless, Jakate et al did not collect data on the efficacy of these drugs for managing migraine.

### 4.4. Gepants and Triptans

Triptans have dominated the market for acute migraine treatment for almost three decades. Thus, the efficacy of this class of medication is well-known. A long-term safety profile with consistent response observations has already been provided. Therefore, there is an important question regarding the use of new drugs for the management of acute migraine. Notably, medication overuse with triptans could be one important reason for developing gepants [49].

A recent systematic review and network meta-analysis compared the benefits of 5-HT1F receptor agonists and CGRP antagonists with triptans for treating acute migraine attacks. They observed that triptans were associated with a higher odds ratio for pain freedom at 2 h compared to lasmiditan, rimegepant, and ubrogepant. There were also no statistically significant differences in the comparisons between lasmiditan, rimegepant, and ubrogepant for pain freedom and pain relief at 2 h. Interestingly, lasmiditan was associated with a higher risk of adverse events than CGRP antagonists [48]. Therefore, CGRP antagonists are a possible “drug of choice” for treating acute migraine in individuals with a significant history of cardiovascular disease [59].

The assessment of efficacy for the concomitant use of triptans and CGRP antagonists is critical, given the high percentage of patients requiring combined therapy for remission of headaches. In this context, a study of hemodynamic effects and pharmacokinetic interactions during concomitant use of rimegepant and sumatriptan in healthy adults was performed [60]. It showed no hemodynamic or pharmacokinetic interaction between these drugs and that this association was safe and well-tolerated [61].

It is worth mentioning that rimegepant is effective for the acute treatment of migraine in subjects who had failed triptans; consistency was observed regardless of the number of triptan failures [62]. However, a good response to triptans is a predictive factor for a positive response to monoclonal antibodies that block CGRP [63].

### 4.5. Acute and Preventive Treatment of Migraine

In pharmacology, the development of new chemical compounds is associated with estimations in the pharmacokinetics and pharmacodynamics of the drug. However, it is challenging to provide strong evidence of these simulations without clinical trials. In this context, rimegepant was observed in the first clinical trials to have a long-term effect of pain freedom at 2 hours that was maintained for 24 hours. After these initial observations, the studies increased the time to 48 hours, and it was noted that an important percentage of patients on rimegepant, compared to the placebo, had pain freedom (Figure 3).

The Croop et al trial compared the 12-week administration of rimegepant 75 mg (n = 373 subjects) or the placebo (n = 374) every other day with the primary endpoint to assess the change from the baseline in the mean monthly migraine days in the last four weeks of the double-blind phase. They observed that, taken every other day, rimegepant was effective for preventive treatment of migraine, and no severe or unexpected safety issues were noted [43].

Popoff et al., systematically evaluated the relative efficacy of rimegepant, atogepant, and monoclonal antibody (erenumab, galcanezumab, eptinezumab, and fremanezumab) treatments for the prevention of migraine. All of these drugs were effective treatment options for the prevention of migraine when compared to the placebo. They did not show significant differences when compared to one another [64].

### 4.6. Future Directions

Further studies are needed for the pharmacological properties, including pharmacodynamics, related to gepants. No significant percentage of side effects was observed within one year, but the evaluation of long-term adverse events of these drugs should be performed, since CGRP receptors are found throughout the body. Thus, the unknown side effects of other systems besides the central nervous system are still an important concern for some patients. For example, there is a theoretical possibility of vascular hemodynamic impairment, leading to chronic cardiovascular and neurovascular damage.

Comparative studies among the medications inside this class and with triptans are mandatory for improving clinical practice. This information will guide clinicians regarding the drugs of choice based on patients’ comorbidities. Clinical trials with gepants are being designed for migraine during pregnancy and pediatric migraine. The spectrum of drugs available for these two groups of individuals is narrow regarding concerns related to serious adverse events.

Another fact that should be addressed in future studies is the inclusion of the elderly population. Most clinical trials have exclusion criteria of about 65–75 years. Considering the increasing aging population in the US and the prevalence of migraine after the sixth decade of life, potential drug targets in this population of the public need further studying.

## 5. Conclusions

Gepants were the first oral agents specifically designed to prevent migraine. The new generation of CGRP antagonists is effective for the acute and preventive treatment of migraine. In acute migraine therapy, the second generation of gepants showed slightly lower potency when compared to triptans, but the side effects were minor without significant central nervous system events. There are still concerns regarding the long-term side effects of these medications, including vascular hemodynamic impairment. Meanwhile, careful pharmacovigilance and safety monitoring should be performed when using gepants in clinical practice.

## Figures and Tables

**Figure 1 brainsci-12-01612-f001:**
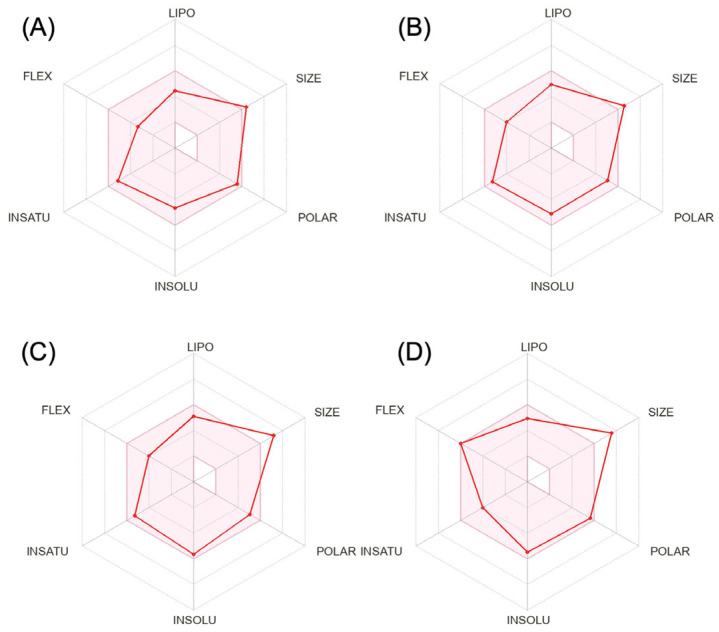
Physicochemical properties of rimegepant (**A**), ubrogepant (**B**), atogepant (**C**), and zavegepant (**D**).

**Figure 2 brainsci-12-01612-f002:**
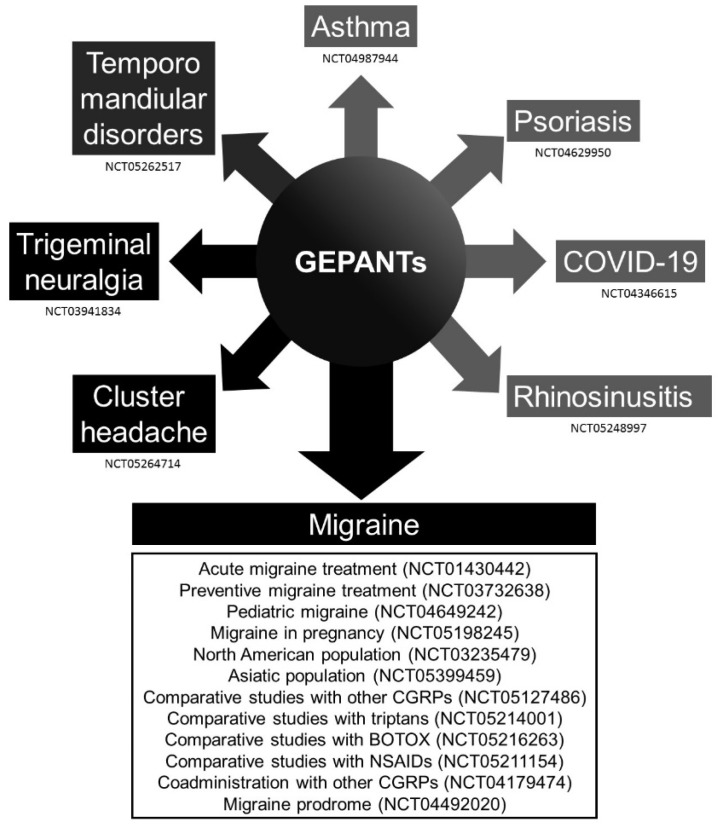
Schematic diagram of the conditions assessed in the clinical trials registered at “ClinicalTrials.gov” with rimegepant, ubrogepant, atogepant, and zavegepant. The colors are graded due to possible different mechanisms related to the calcitonin gene-related peptide (CGRP) pathway.

**Figure 3 brainsci-12-01612-f003:**
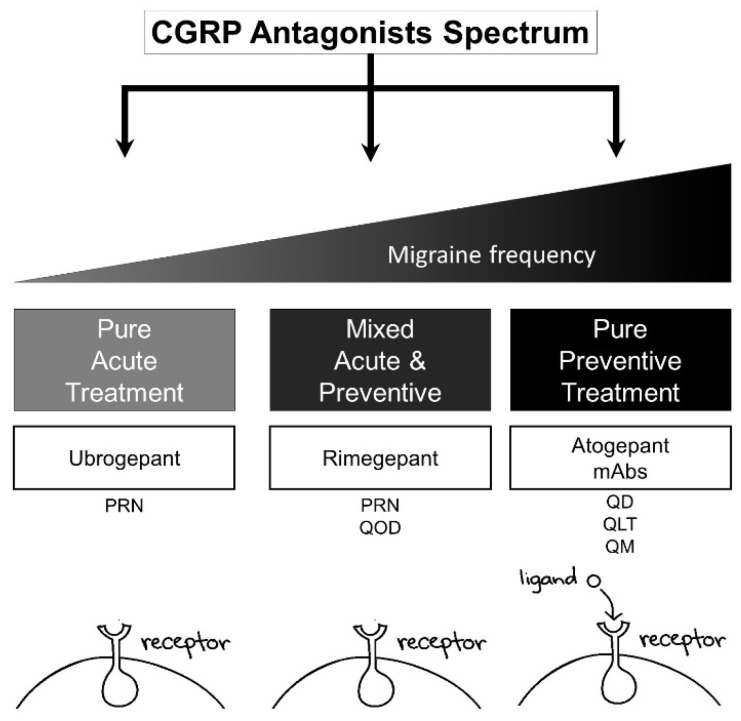
Spectrum of calcitonin gene-related peptide (CGRP) medications in clinical practice. The increase in migraine frequency per month is related to preventive therapy. Apparently, treatment is associated with a drug target that could be a receptor or ligand. Receptor target: erenumab, ubrogepant, rimegepant, and atogepant. Ligand target: fremanezumab, galcanezumab, and eptinezumab. PO: QoD, every other day; QD, daily; PRN, as needed; QLT, quarterly; QM, monthly.

**Table 1 brainsci-12-01612-t001:** Comparative analyses of pharmacokinetics, dosage, and formulations of rimegepant, ubrogepant, and atogepant.

Drugs	Rimegepant	Ubrogepant	Atogepant
Other names	BHV-3000, BMS-927711	MK-1602	AGN-241689, MK-8031
Trade name	Nurtec^®^, Nurtec ODT®, Vydura®	Ubrelvy^®^	Qulipta^®^
Marketed	Biohaven Pharmaceuticals, Pfizer	Allergan Pharmaceuticals	Allergan Pharmaceuticals
FDA approval	February 2020 (acute); May 2021 (prevention)	December 2019	September 2021
Molecule	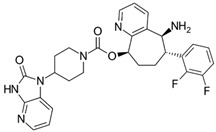	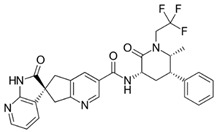	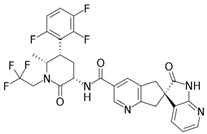
Indication	Acute treatment of migraine with or without aura in adults. Preventive treatment of episodic migraine in adults	Acute treatment of migraine with or without aura in adults	Preventive treatment of episodic migraine in adults
Dose	75 mg PO as needed. Not to exceed 75 mg/ 24 h. Safety of >18 doses/30 days has not been established	50 or 100 mg PO as needed. If needed, may take second dose at least 2 hrs after initial dose. Not to exceed 200 mg/24 h. Safety of treating >8 migraines/30 days has not been established	10, 30, or 60 mg PO
Dose adjustments	Renal impairment (CrCl < 15 mL/min); hepatic impairment (Child–Pugh > C); drug interactions	Renal impairment (CrCl < 30 mL/min); hepatic impairment (Child–Pugh ≥ C); drug interactions	Renal impairment (CrCl < 30 mL/min); hepatic impairment (Child–Pugh ≥ C); drug interactions
Route of administration	Oral	Oral	Oral
Contraindications	History of hypersensitivity	Concomitant use with strong CYP3A4 inhibitors	None
Adverse effects	Nausea (2%), hypersensitivity (including delayed)	Nausea (2–4%), somnolence (2–3%), dry mouth (2%)	Nausea (5–9%), constipation (6%), somnolence (4–6%), elevated AST/ALT (1%)
Pregnancy and lactation	Data not available	Data not available	Data not available
Pediatric	Data not available	Data not available	Data not available
Considerations	Orally disintegrating tablet. It can be swallowed without additional liquid	High-fat meal delayed plasma concentration	High-fat meal effect was not significant.

**Table 2 brainsci-12-01612-t002:** Clinical trials of gepants indicated for acute migraine therapy.

Drug	Reference, Year (NCT)	Groups Gepants (mg) & Placebo	Number of Participants Analyzed (Completed)	Dropout Rate (Number of Participants)	At 2 h, Pain Freedom (%); MBS (%)	AE (%)	SAE (n)
Rimegepant	2012 (NCT01430442)	10	72	15.3 (13)	19.7	5.5	0
		25	62	8.8 (6)	19.7	0.0	0
		75	86	5.5 (5)	31.4	3.4	0
		150	86	4.4 (4)	32.9	3.4	3
		300	112	7.4 (9)	29.7	5.3	0
		600	84	8.7 (8)	24.4	8.3	0
		Placebo	210	8.3 (19)	15.3	2.3	0
	Lipton et al., 2018 (Study 301, NCT03235479)	75	541	7.0 (41)	19.2; 36.6	12.6	2
		Placebo	540	6.9 (40)	14; 27.7	10.7	1
	Lipton et al., 2019 (Study 302, NCT03237845)	75	538	9.4 (56)	19.6; 37.6	17.1	1
		Placebo	542	8.4 (50)	12; 25.4	14.2	2
	Croop et al., 2019 (Study 303, NCT03461757)	75	679	7.2 (53)	21; 35	13.2	0
		Placebo	689	6.1 (45)	11; 27	10.5	0
Ubrogepant	2016 (NCT01657370)	1	28	12.5 (4)	0.0; 42.9 *	21.4	0
		10	25	21.9 (7)	3.8; 30.8 *	7.6	0
		25	28	15.2 (5)	17.9; 42.9 *	35.7	0
		50	27	20.6 (7)	28.6; 57.1 *	25.0	0
		100	27	12.9 (4)	11.1; 44.4 *	25.9	0
		Placebo	27	18.2 (6)	0.0; 28.6 *	21.4	0
	Voss et al., 2016 (NCT01613248)	1	104	24.6 (34)	5.6; 45.8	20.5	0
		10	106	23.7 (33)	14.8; 49.1	12.0	0
		25	101	27.3 (38)	21.4; 55.3	14.4	0
		50	104	25.2 (35)	21.0; 56.2	13.2	1
		100	102	27.1 (38)	25.5; 60.8	16.6	0
		Placebo	110	20.9 (29)	8.9; 42	13.2	0
	Dodick et al., 2019 (Achieve I, NCT02828020)	50	457	17.8 (99)	19.2; 38.6	9.4	3
		100	479	14 (78)	21.2; 37.7	16.3	2
		Placebo	479	14.3 (80)	11.8; 27.8	12.8	0
	Lipton et al., 2019 (Achieve II, NCT02867709)	25	474	15.5 (87)	20.7; 34.1	9.2	1
		50	487	13.3 (75)	21.8; 38.9	12.9	0
		Placebo	493	12.4 (70)	14.3; 27.4	10.2	0
Zavegepant	Croop et al., 2021 (NCT03872453)	5	387	NA	NA	NA	NA
		10	391	NA	22.5; 41.9	13.5	NA
		20	402	NA	23.1; 42.5	16.1	NA
		Placebo	401	AN	15.5; 33.7	3.5	NA

Abbreviations: AE, adverse event; MBS, most bothersome symptom; NA, not available/ not reported; NCT, clinical trial identifier number; SAE, serious adverse event. * MBS was not utilized in this study, and we presented data on phonophobia at 2 h. SAE reported: (a) rimegepant: stress cardiomyopathy, pneumonia, post-lumbar puncture syndrome, acute respiratory failure, pulmonary embolism, back pain; (b) ubrogepant: myoclonus, pericardial effusion, seizure, spontaneous abortion.

**Table 3 brainsci-12-01612-t003:** Clinical trials of gepants indicated for preventive migraine therapy.

Drug	Reference, Year (NCT)	Groups Gepants (mg) & Placebo	Number of Participants Analyzed (Completed)	Dropout Rate (Number of Participants)	Change in Monthly Migraine Days	AE (%)	SAE (n)
Atogepant	Goadsby et al., 2020 (NCT02848326)	10 QD	80	14.9 (14)	−4.0	18.2	1
		30 QD	149	19.5 (36)	−3.7	25.6	2
		60 QD	164	12.3 (23)	−3.5	27.9	2
		30 BD	70	21.3 (19)	−4.2	24.4	0
		60 BD	73	21.5 (20)	−4.1	28.5	0
		Placebo	148	20.4 (38)	−2.8	19.3	2
	Schwedt et al., 2022 (ADVANCE, NCT03777059)	10 QD	193	13.1 (29)	−3.6	15.3	2
		30 QD	207	10.0 (23)	−3.8	16.2	0
		60 QD	204	13.2 (31)	−4.2	13.8	0
		Placebo	201	9.9 (22)	−2.4	6.7	2
Rimegepant	Croop et al., 2021 (NCT03732638)	75 EOD	316	14.6 (54)	−4.3	0	3
		Placebo	310	16.4 (61)	−3.5	0	0

Abbreviations: AE, adverse event; EOD, every other day; SAE, serious adverse event. SAE reported: (a) atogepant: cholecystitis, ureteritis, worsening migraine episodes, optic neuritis, asthma; (b) rimegepant: gastroenteritis, malignant melanoma, suicide attempt.

## Data Availability

Not applicable.

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
