# Peer review of "Gepants for Acute and Preventive Migraine Treatment: A Narrative Review"

_brainsci, 2022, doi:10.3390/brainsci12121612_

Round 1

Reviewer 1 Report

Thi is a well conceived review focusing on migraine acute and preventing treatment with gepants. The Introduction is well synthetized and adequately referenced. Table 1 and Figure 1 appears to be useful for quick evaluation of pharmacological properties. The collection of RCT data in Table 2 and Table 3 is comprehensive and useful for consultation. To sum up, this paper would be valuable for phisicians involved in migraine management.

Author Response

We would like to thank the Reviewer for taking the time to improve the quality of our manuscript. Also, we would like to reinforce that the manuscript was primarily done to summarize the current evidence of gepants in the treatment of acute and preventive management of migraine.

Reviewer 2 Report

This is a summary of the clinical application of calcitonin gene-related peptide (CGRP) antagonist gepants. The authors focus on the chemical properties of second- and third-generation gepants and their therapeutic and preventive effects on acute migraine. In addition, the authors compared the side effects of these CGRP antagonists. This review provides some evidence and basis for the clinical treatment and prevention of migraine, however, the lack of some important statements or concluding points in this manuscript makes the reader appear somewhat difficult and confused. The authors need to address the following questions before the manuscript is ready for publication.

Questions

1.       The authors summarized in the abstract that "no increased mortality risk was observed associated with the second and third-generation gepants", however, the authors didn’t further explain why this claim was reached in the manuscript.

2.       Many references in the manuscript are missing.

3.       What is the purpose of the author listing table 1? It’s better to give a clear statement about the differences between those Gepants. Also, what is the half-life of the drug?

4.       The authors indicate "These properties help identify compounds suitable for oral use", on what basis did the authors make this point?

5.       What are the authors trying to tell readers in Table 2? It's better for authors to give clear arguments.

6.       Why do the authors think "atogepant's affinity at the CGRP receptor in humans is higher than that of ubrogepant"? Are there literature references to support this argument?

7.       The manuscript describes “atogepant has the highest percentage of side effects, including serious adverse events”, what kind of serious side effects? Are there references to support this?

Author Response

REVIEWER 1

This is a summary of the clinical application of calcitonin gene-related peptide (CGRP) antagonist gepants. The authors focus on the chemical properties of second- and third-generation gepants and their therapeutic and preventive effects on acute migraine. In addition, the authors compared the side effects of these CGRP antagonists. This review provides some evidence and basis for the clinical treatment and prevention of migraine, however, the lack of some important statements or concluding points in this manuscript makes the reader appear somewhat difficult and confused. The authors need to address the following questions before the manuscript is ready for publication.

Questions

  1. The authors summarized in the abstract that “no increased mortality risk was observed associated with the second and third-generation gepants”, however, the authors didn’t further explain why this claim was reached in the manuscript.

Authors: We appreciate the Reviewer’s comment. The abstract statement is regarding the first paragraph of section “3. Clinical trials related to Gepants.” In the clinical trials already published with different second and third-generation gepants, no increased mortality risk was observed compared to placebo. The authors believe that the main focus of study for future trials should be individuals with previous cardiovascular risk factors. There are no data regarding this specific group of individuals. Also, the long-term cardiovascular outcomes were not studied.

  1. Many references in the manuscript are missing.

Authors: We revised the manuscript to address this comment. The main manuscript has 64 references. In the supplementary material, there is a description of the data, and some references for further analysis are provided. However, we did not include these in the manuscript because they do not provide sufficient evidence for being cited. We would like to request the Reviewer for specific references that the authors could cite.  

  1. What is the purpose of the author listing table 1? It’s better to give a clear statement about the differences between those Gepants. Also, what is the half-life of the drug?

Authors: The purpose of Table 1 is to provide enough knowledge for the reader to understand the main points that will be further discussed. The authors would like to say that no similar table was found in the literature. Also, “Supplementary material 2” describes the specific pharmacokinetic characteristics of the studied drugs. Moreover, most of the information provided was found in different studies, and some pharmacological features were calculated using the SwissADME tool.

  1. The authors indicate “These properties help identify compounds suitable for oral use”, on what basis did the authors make this point?

Authors: The statement “These properties help identify compounds suitable for oral use” is based on the SwissADME tool parameters. This software asses six physicochemical properties (lipophilicity, size, polarity, solubility, flexibility, and saturation). In this context, optimal physicochemical properties are directly related to oral bioavailability. This information is described in the reference cited in the manuscript.

Daina A, Michielin O, Zoete V. SwissADME: a free web tool to evaluate pharmacokinetics, drug-likeness and medicinal chemistry friendliness of small molecules. Sci Rep. 2017 Mar 3;7:42717. doi: 10.1038/srep42717. PMID: 28256516; PMCID: PMC5335600.

  1. What are the authors trying to tell readers in Table 2? It’s better for authors to give clear arguments.

Authors: Table 2 is provided for further understanding of the clinical trials by the reader. It is the best way to present the main finding of the studies with gepants. The other format would be a complete description, but it would be difficult comprehension. The present manuscript is the first to assess the dropout rate and the distinction between “adverse event” and “serious adverse event.” We observed an important dropout rate with the clinical trials of ubrogepant.

  1. Why do the authors think “atogepant’s affinity at the CGRP receptor in humans is higher than that of ubrogepant”? Are there literature references to support this argument?

Authors: We would like to highlight the importance of this question by the Reviewer. The affinity of gepants to CGRP receptors was reported in the literature with different results. “Supplementary material 2” describe the possible affinity of some gepants to the ‘Human CGRP receptor,’ in which the affinity (pM) was calculated based on animal models. However, the affinity to CGRP receptors was directly assessed in human subjects in the study mentioned (Strassman). Strassman et al. observed that the numbers reported related to atogepant affinity to CGRP receptors are underestimated. And there is a significant difference between gepants and CGPR’s affinity when comparing animal models and human subjects.

Strassman AM, Melo-Carrillo A, Houle TT, Adams A, Brin MF, Burstein R. Atogepant - an orally-administered CGRP antagonist - attenuates activation of meningeal nociceptors by CSD. Cephalalgia. 2022 Aug;42(9):933-943. doi: 10.1177/03331024221083544. Epub 2022 Mar 25. PMID: 35332801; PMCID: PMC9329220.

Moore E, Fraley ME, Bell IM, Burgey CS, White RB, Li CC, Regan CP, Danziger A, Stranieri Michener M, Hostetler E, Banerjee P, Salvatore C. Characterization of Ubrogepant: A Potent and Selective Antagonist of the Human Calcitonin Gene‒Related Peptide Receptor. J Pharmacol Exp Ther. 2020 Jan 28:jpet.119.261065. doi: 10.1124/jpet.119.261065. Epub ahead of print. Erratum in: J Pharmacol Exp Ther. 2021 Jan;376(1):147. PMID: 31992609.

  1. The manuscript describes “atogepant has the highest percentage of side effects, including serious adverse events”, what kind of serious side effects? Are there references to support this?

Authors: We appreciate the Reviewer’s query. There was a higher percentage of side effects with atogepant compared to other gepants. But, no network meta-analysis or head-to-head studies reveal this finding. It is mainly based on the rates of the clinical trials. The adverse events classified as serious by the researchers in the atogepant’s clinical trials were ureteritis, worsening migraine episodes, optic neuritis, and asthma.

Switzer MP, Robinson JE, Joyner KR, Morgan KW. Atogepant for the prevention of episodic migraine in adults. SAGE Open Med. 2022 Oct 6;10:20503121221128688. doi: 10.1177/20503121221128688. PMID: 36226229; PMCID: PMC9549103.

Reviewer 3 Report

Thank you for this very nice review about gepants. My comments:

-L30: “The incidence of..   ….100,000 people in 2017” or you can switch this sentence with the next one. Make it clear that the reference is the same [2]

- L65: maybe only a view words about the migraine-inducing effect of CGPR and the action of gepants in relation with the BBB. It is a hot topic and it will be discussed in future for sure, so it would be great if you could include it in your wonderful work.

- table 2: where are the references for the 2012 trial of Rimegepant and for the first 2016 trial of ubrogepant

- Figure 2: I don’t understand this figure. Are these all clinical studies with gepants against different health conditions, right? And I can’t see colors (in my copy) only grayscale. So, what you're saying about “possible mechanisms related to CGRP and vascular and inflammatory pathways” needs more analysis and explanation and I'd suggest having what you want to say in a text rather than a figure.

- figure 3: very nice figure, needs only a small correction: “Rimegepant” in place of “Remigepant”

Author Response

REVIEWER 2

Thank you for this very nice review about gepants. My comments:

-L30: “The incidence of..   ….100,000 people in 2017” or you can switch this sentence with the next one. Make it clear that the reference is the same [2]

Authors: We appreciate the Reviewer’s correction. To address this comment, we changed the sentence. “The incidence of migraine headaches is 1,722 cases per 100,000 people in 2017.”

- L65: maybe only a view words about the migraine-inducing effect of CGPR and the action of gepants in relation with the BBB. It is a hot topic and it will be discussed in future for sure, so it would be great if you could include it in your wonderful work.

Authors: We would like to request the Reviewer a reference or phrase to include. The current format has the following description: “Lassen et al. infused CGRP in individuals with migraine to elucidate the mechanism of CGRP and migraine. They observed that intravenous administration of CGRP causes migraine in migraineurs.”

- table 2: where are the references for the 2012 trial of Rimegepant and for the first 2016 trial of ubrogepant

Authors: The authors could not find published results of these clinical trials in the literature. Therefore, the authors directly extracted from “ClinicalTrials.gov.”

“clinical trials related to rimegepant, ubrogepant, atogepant, and zavegepant registered in the ‘ClinicalTrials.gov’”

- Figure 2: I don’t understand this figure. Are these all clinical studies with gepants against different health conditions, right? And I can’t see colors (in my copy) only grayscale. So, what you’re saying about “possible mechanisms related to CGRP and vascular and inflammatory pathways” needs more analysis and explanation and I’d suggest having what you want to say in a text rather than a figure.

Authors: The figure was designed to describe gepant’s studies in different conditions. There is a grayscale describing possible main vascular to main inflammatory pathways. These affirmations are based on the registration of the trials in “ClinicalTrials.gov.” It is a complex figure and involves all the clinical trials. The main aim of the manuscript is related to migraine disorders. We believe the present format already has a full explanation for acute and preventive migraine treatment.

- figure 3: very nice figure, needs only a small correction: “Rimegepant” in place of “Remigepant”

Authors: We would like to thank the Reviewer for this grammatical error. We corrected the figure.

Reviewer 4 Report

Great paper! 

1. In the first paragraph you cite costs per different types of visits. Are you trying to point out that it is cheaper to have effective acute treatment at home and avoid ED visits?

I think supplementary material 1 is labeled incorrectly- it was referred to in the introduction but I'm not sure what it was supposed to be

2. In table 1 you state that the safety of treating more than 8 migraines in 30 days hasn't been established. I would recommend looking at Pub Med ID 31899602, specifically their Trial B. Daily use of ubrogepant 150mg was not associated with any significant adverse effects (granted, this was not used to treat migraine attacks daily)

You mention that rimegepant data was reanalyzyed, and that this reanalysis spurred the development of the second generation of gepants, which include rimegepant. This is unclear. 

I do not think your reference #26 is correctly applied (using a review paper rather than the original)

In your discussion of rimegepant's delivery system, you mention that this improves compliance due to avoiding nausea, without reference. I would suggest adding a reference to back up this statement or qualify that this is the opinion of the authors (as my clinical experience seems to be a bit different than yours)

Figure 2 should have text that explains the acronyms used in the figure

3. Your paragraph covering rimegepant discusses the dose-finding trials, and briefly mentions the adverse effects. The section on ubrogepant has a very different tone and spends a much larger portion on adverse effects, and hypothesizes a mechanism (despite earlier statements in your paper suggesting that adverse effects were less likely to be related to the medications). This seems very biased.

4.1 It may be more helpful to flush this section out a bit more- when mentioning that one medication was more efficacious it may be helpful to add in the actual outcomes (pain freedom percentage, etc), so that the reader can get a better understanding of clinical vs statistically significant changes, and the changes in effect size

It may be helpful to discuss risk of Medication overuse with triptans as another reason to think about the use of gepants

Author Response

We appreciate the Reviewer's comments for the improvement of the manuscript quality.

Round 2

Reviewer 2 Report

The authors have addressed all my questions. I have no further concerns about the manuscript. 

Author Response

We would like to thank the Reviewer for taking the time to improve the quality of our manuscript.

Reviewer 4 Report

I appreciate the hard work, and look forward to seeing it published